# Solubility of Hydrogen in a WMoTaNbV High-Entropy Alloy

**DOI:** 10.3390/ma17112574

**Published:** 2024-05-27

**Authors:** Anna Liski, Tomi Vuoriheimo, Jesper Byggmästar, Kenichiro Mizohata, Kalle Heinola, Tommy Ahlgren, Ko-Kai Tseng, Ting-En Shen, Che-Wei Tsai, Jien-Wei Yeh, Kai Nordlund, Flyura Djurabekova, Filip Tuomisto

**Affiliations:** 1Department of Physics and Helsinki Institute of Physics, University of Helsinki, 00014 Helsinki, Finland; jesper.byggmastar@helsinki.fi (J.B.); kenichiro.mizohata@helsinki.fi (K.M.); tommy.ahlgren@helsinki.fi (T.A.); kai.nordlund@helsinki.fi (K.N.); flyura.djurabekova@helsinki.fi (F.D.); filip.tuomisto@helsinki.fi (F.T.); 2Department of Nuclear Sciences and Applications, International Atomic Energy Agency, Vienna International Centre, 1220 Wien, Austria; k.heinola@iaea.org; 3Department of Materials Science and Engineering, National Tsing Hua University, Hsinchu 300044, Taiwan; zouts2@gmail.com (K.-K.T.); timoshen1203@gmail.com (T.-E.S.); chewei@mx.nthu.edu.tw (C.-W.T.); jwyeh@mx.nthu.edu.tw (J.-W.Y.); 4High Entropy Materials Center, National Tsing Hua University, Hsinchu 300044, Taiwan

**Keywords:** high entropy alloy, metals, hydrogen, deuterium, solution enthalpy, activation energy, elastic recoil, density functional theory

## Abstract

The WMoTaNbV alloy has shown promise for applications as a solid state hydrogen storage material. It absorbs significant quantities of H directly from the atmosphere, trapping it with high energy. In this work, the dynamics of the absorption of hydrogen isotopes are studied by determining the activation energy for the solubility and the solution enthalpy of H in the WMoTaNbV alloy. The activation energy was studied by heating samples in a H atmosphere at temperatures ranging from 20 °C to 400 °C and comparing the amounts of absorbed H. The solution activation energy $E_{A}$ of H was determined to be $E_{A}=0.22\pm 0.02$ eV (21.2 ± 1.9 kJ/mol). The performed density functional theory calculations revealed that the neighbouring host atoms strongly influenced the solution enthalpy, leading to a range of theoretical values from −0.40 eV to 0.29 eV (−38.6 kJ/mol to 28.0 kJ/mol).

## 1. Introduction

The hydrogen-induced embrittlement in metals, mainly steels, was one of the early focal points in metal–hydrogen research [1]. The interest has progressively widened to bcc-structured transition metals, following the rapid increase of their usage in various technological and engineering applications, such as structural materials [2]. These metals facilitate unique interactions with freely migrating hydrogen atoms, many of which are related to its small atomic size [3]. Hydrogen is found to occupy the tetrahedral site in the bcc lattice and is able to diffuse with exceptionally high mobility amidst the lattice atoms [4]. Its diffusivity values in bcc transition metals are of the order of ions in aqueous solutions, allowing metal–hydrogen systems to find their thermodynamic equilibrium rapidly, even at room temperature [3]. The ability to absorb and release large amounts of hydrogen is one of the most critical characteristics of metals and metallic alloys considered to be hydrogen storage materials [5]. These materials are highly sought after for hydrogen fueled applications such as heat pumps, metal hydride batteries, and polymer electrolyte fuel cells [6,7]. Considering practical usage, hydrogen-absorbing alloys have primarily consisted of simple intermetallic compounds such as ${\mathrm{LaNi}}_{5}$, TiV, TiFe, and ${\mathrm{TiMn}}_{1.5}$ [6,8]. A step towards more complex structures was presented by Okada et al., who introduced Ti-based ternary bcc alloys for hydrogen storage in the early 2000s [7]. The ternary alloys allow for easier activation and an increased reversible storage capacity for hydrogen compared with binary alloys [9]. Hydrogen storage materials have since been evolving towards a higher complexity with the recent discovery of hydrogen absorbing high-entropy alloys (HEAs), which allow for exceptionally high hydrogen storage levels, reaching hydrogen-to-metal (H/M) ratios up to 2.5 [10]. The substantial hydrogen storage capacity is attributed to the inherent characteristics of HEA materials. The complexity of the lattice structure allows them to absorb hydrogen on multiple different trapping sites, thus increasing the H/M ratio beyond the pure element average [10].

HEAs are built from multiple main components in equal or near equal proportions [11]. The constituent elements are randomly ordered in the lattice forming a singular solid solution phase, at least at high temperatures. The uniform phase is a defining feature of high-entropy alloys, influencing some of the material’s distinct properties. The ability to achieve high H/M ratios is founded on the irregularity of the lattice structure, which limits the mobility of interstitials and creates additional trapping sites to be occupied by hydrogen atoms [10]. The exceptional mechanical properties of the high-entropy alloys can also be attributed to the synergistic inter-element interactions [11,12]. Bcc-structured high-entropy alloys, such as WMoTaNbV, outperform the traditional alloys and many of the high-performance superalloys (Inconel, Waspaloy) when it comes to high temperature applications [13]. They demonstrate an exceptional mechanical durability, high irradiation tolerance, and good thermal conductivity, making them auspicious materials for extreme environment applications (e.g., aerospace [14], marine [15], and nuclear [16]).

The prospective for industrial applications of high-entropy alloys underline the importance of gaining qualitative knowledge about the processes influencing the absorption of hydrogen. In previous research, a number of V-rich HEAs have been observed to absorb hydrogen, this includes the WMoTaNbV alloy studied in this work [17,18]. The WMoTaNbV efficiently traps hydrogen, with an energy of ∼1.7 eV [18]. The exchange between the H and D isotopes, however, occurs at temperatures related to much lower energies, indicating a similarity between the trapping of two isotopes H and D [19]. This work confirms the hypothesis of the similar trapping mechanism between isotopes and further investigates the dynamics of hydrogen trapping.

## 2. Experimental Methods

### 2.1. Sample Processing

The equimolar refractory alloy WMoTaNbV was manufactured at the High Entropy Materials Center of National Tsing Hua University by vacuum arc melting [20]. The casting process resulted in an alloy with a single bcc crystalline phase. The alloy had a fine grain structure, with the average grain size in the order of ∼80 μm, although with significant variations between the sizes of the individual grains [21]. Additionally, the grains had a dendritic shape and their elemental composition showed a microscale segregation of the constituent elements. The grain boundaries were enriched with Mo, V, and Nb and the centers of the grain with W [22]. The concentrations of components in the alloy were reported to be Mo (20.7 at.%), Nb (20.3 at.%), Ta (20.4 at.%), V (19.2 at.%), and W (19.4 at.%), as determined by energy dispersive spectroscopy (EDS) and its bcc structure was confirmed by X-ray diffraction measurement (XRD) performed by the manufacturer [20]. The samples were received cut with dimensions 5 × 5 × 1 ${\mathrm{mm}}^{3}$. Each piece of sample material was mechanically polished by diamond suspension by lowering the polishing particle size progressively down to 50 nm. Casting of the alloy may introduce impurities of C, O, and N in the lattice. Annealing of the samples allows for the desorption of N, O, and H along with the partial removal of C through desorption as CO and ${\mathrm{CO}}_{2}$. The polished samples were annealed in a quartz tube oven with a vacuum of $10^{-7}$ mbar at 1000 °C for 2 h to reduce the amount of impurities. The annealing temperature was sufficiently low so as not to induce recrystallization in the samples [21]. Subsequently, the samples were implanted with deuterium (D) as molecular $D_{2}^{+}$ ions using 30 keV/D implantation energy. Energetic $D^{+}$ implantations were used to monitor the probability of the previously observed isotopic exchange of hydrogen isotopes H and D [19] and to verify the similarity in the trapping mechanism of these isotopes. The fluence of the implantation was $5.8\times 10^{16}$ D/cm^2^. The implantation process is detailed in Ref. [19]. The implanted WMoTaNbV samples were annealed in a $H_{2}$ atmosphere with a quartz tube oven for 4 h while maintaining the $H_{2}$ gas pressure at 1 bar. The annealing time was chosen based on previous HEA isotopic exchange work [19]. Gas flow within chamber was kept constant throughout the $H_{2}$ atmosphere annealing. The annealing temperatures were 20 °C, 100 °C, 200 °C, 300 °C, and 400 °C.

### 2.2. Elastic Recoil Detection Analysis

The concentration depth profiles of the retained H and D were measured with Elastic Recoil Detection Analysis (ERDA) using 24 MeV ^28^Si^5+^ ions at an incident angle of 30°. The detector was set to a 45° angle. The configuration achieved a probing depth up to 400 nm from the surface.Heavier particles were filtered out with 4 μm havar foil at the entrance of the detector. The detector had a nominal energy resolution of 15 keV. It was positioned at a distance of 50 mm from the sample and its solid angle was limited to 2 × 7 mm^2^ with a collimator. A metal net around sample holder, set to −300 V, was used to suppress the secondary electrons for the more accurate beam current and fluence measurement. The measurement configuration was selected to maximize depth resolution and to prevent the overlap of H and D signals. The ERDA spectra were analyzed for both H and D signals. The raw spectra are displayed in Figure 1. The surface H, which is visible in the ERDA data, does not contribute to thermodynamic equilibrium of solute H. Therefore, the surface H signal is omitted from the analysis by fitting the H bulk profile. As H and D have similar chemical properties and hence trapping features, the retained amount of implanted D contributes to the total amount of subsequently dissolved H in the WMoTaNbV alloy by occupying traps otherwise available for H. An unimplanted sample was used as the background reference.

## 3. Modeling Methods

The Vienna Ab Initio Simulation Package (VASP) [23,24] with the PBE GGA exchange-correlation functional [25] was used to perform the density functional theory (DFT) calculations. The k-point grid was 4×4×4 and the plane-wave energy cutoff was 500 eV. The projector augmented wave potentials [26] included semi-core electrons for all of the elements. We used supercells of WMoTaNbV containing 128 atoms in random order. Ten bulk systems with different random orders were prepared and relaxed to zero pressure in the conjugate gradient energy minimizations. In each relaxed bulk system, five different cells with one randomly placed H atom each were created and minimized (at constant volume). During relaxation, the H atom moved to a nearby stable tetrahedral site in all of the cases. The theoretical solution enthalpy was obtained as
(1)$$E_{s}=E_{\mathrm{bulk}+H}-E_{\mathrm{bulk}}-E_{H}.$$

Here, $E_{\mathrm{bulk}+H}$ and $E_{\mathrm{bulk}}$ are the relaxed bulk energies with and without one H atom and $E_{H}$ is the reference energy of H obtained from the total energy of the $H_{2}$ dimer as −3.3855 eV/atom. We also calculated the solution enthalpies in all five pure metals for comparison.

## 4. Results

### 4.1. Activation Energy of H Solution

The absorption of hydrogen into metals follows a three-step process consisting of adsorption, dissociation, and dissolution (see Figure 2). The balance of the dissolved amount was governed by the fraction of the partial pressure of hydrogen in the gas $p=p_{H_{2}}/p_{0}$ and the ratio of atoms H/M in the metal, together termed Sievert’s constant. Sievert’s law, which is used for predicting the solubility of gases in metals and describes the thermodynamic equilibrium of the hydrogen–metal system is expressed as
(2)$$H/M=\beta \sqrt{p}\exp\left(\frac{\mu_{H_{2}}^{{}^{\circ}}-2\mu_{H}^{{}^{\circ}}}{2k_{B}T}\right)$$
where β is the ratio of interstitial sites to metal atoms [27]. Hydrogen is occupying tetrahedral sites in the WMoTaNbV alloy, where β=6 [27], as the total number of possible sites per unit cell is divided by the number of its atoms. The chemical potentials, $\mu_{H_{2}}^{{}^{\circ}}$ for the surrounding $H_{2}$ gas and $\mu_{H}^{{}^{\circ}}$ for the H atoms dissolved in the metal, are the same in the equilibrium conditions, setting the Gibbs free energy $\Delta G^{{}^{\circ}}$ to zero [27].
(3)$$\Delta G^{{}^{\circ}}=\Delta H^{{}^{\circ}}-T\Delta S^{{}^{\circ}}=2\mu_{H}^{{}^{\circ}}-\mu_{H_{2}}^{{}^{\circ}}$$

The $\Delta S^{{}^{\circ}}$ is constant and equivalent to the entropy of gaseous hydrogen [28], while the enthalpy $\Delta H^{{}^{\circ}}$ is determined by the elastic and electronic structure of the metal [27]. Transition metals on the left side of the periodic table of the elements hold strongly negative values of solution enthalpy [29]. The enthalpy flips to the positive side at group VI, making two components of the WMoTaNbV alloy (W and Mo) non-absorbent of hydrogen. The rest of the alloy components (V, Nb, and Ta) are in group V with slightly negative solution enthalpies. They are generally considered as promising materials for hydrogen storage as they are able to absorb large quantities of hydrogen on stable absorption sites [30].

The ERDA spectra were analyzed for bulk concentrations of both isotopes of hydrogen (H and D). Due to the excellent fit of the Arrhenius law in the data, the samples were concluded to have reached an equilibrium state in their hydrogen concentrations. The amounts of absorbed D + H in each sample as measured with ERDA are summarized in Figure 3. An Arrhenius law fit to the results yielded a value for the solution activation energy $E_{A}$ as 0.22±0.02 eV (21.2 ± 1.9 kJ/mol). The results obtained at RT were omitted from the Arrhenius fitting as the temperature was insufficient for a notable increase in absorbed hydrogen concentration. At RT, however, the alloy was observed to absorb hydrogen from the natural atmosphere, which would indicate the hydrogen absorption to be at least partially an exothermic reaction with negative $\Delta G^{{}^{\circ}}$ values.

The ERDA spectra were analyzed for bulk concentrations of both isotopes of hydrogen. H and D had similar chemical properties and their trapping dynamics were differentiated only by the small difference in zero point energies, whereas the trapping mechanisms were the same. This mechanism was validated with a single background sample that was not implanted with D, containing only H as followed by a natural absorption in the atmospheric conditions. The H concentrations in the background sample differed minimally when compared with the D implanted sample that was left in $H_{2}$ atmosphere at room temperature. Room temperature was insufficient to cause notable absorption into the lattice. The equilibrium concentrations of H in the WMoTaNbV alloy increased considerably between $H_{2}$ gas heating temperatures from 100 °C to 200 °C.

### 4.2. H Solution Enthalpy

The solution enthalpy $E_{s}$ of hydrogen in the WMoTaNbV alloy from the DFT calculations range from negative values, −0.40 eV (−38.6 kJ/mol) with a distribution that stretches to 0.29 eV (28.0 kJ/mol), approaching the values of group-VI elements (Mo and W) that have the highest solution enthalpies as pure metals. The distribution is shown in Figure 4a). The configurations of H in WMoTaNbV where the solution enthalpy resulted in slightly positive values had Mo and W as the nearest neighbours more frequently. In contrast, lower values for the solution enthalpy were most commonly found in the configurations where the hydrogen atom was positioned in the vicinity of Ta, Nb, and V. Hence, the elements of the nearest neighbors strongly controlled the solution enthalpy, although not completely, suggesting that the enthalpy is additionally affected by other features of the WMoTaNbV structure, such as possible localized lattice strains around the hydrogen atom. Figure 4b) shows this as the correlation between the solution enthalpies and the number of W and Mo atoms as the nearest neighbors.

Further, the possibility of hydrogen remaining as a molecule in the bulk was examined with the present DFT calculations. It was shown, that placing $H_{2}$ as a molecule within the studied lattice consistently resulted in two separated H atoms due to their strong mutual repulsion. This is in line with numerous studies on H in W, which underline that $H_{2}$ molecules do not form in intact W crystals [31,32,33].

## 5. Discussion

The range of solution enthalpies obtained from the present DFT calculations is consistent with previous calculations of hydrogen dissolution in HEAs. A quaternary refractory alloy, NbMoTaW, yields similar values as obtained in this work, with a marginally broader range from −0.24 eV to 0.51 eV [34]. The ranges for hydrogen solution enthalpies have also been studied in the fcc structured alloys FeCrCoNi and FeCuCrMnMo, considering both the tetrahedral and octahedral positioning of the hydrogen atom in the lattice. On the basis of the results, Ren et al. proposed a correlation between the variance of the enthalpy ranges and the degree of lattice distortion [34]. Similar considerations were created by Sahlberg et al. [10], who suggested that the complex lattice structure has an effect on hydrogen trapping by providing a variety of sites with different trapping energies. Lattice strains create irregularities in the structure, some of which can translate to changes in interactions with hydrogen. The DFT obtained negative values for solution energies coincide well with the currently observed absorption of hydrogen from the atmosphere at RT, yet the absorption is likely limited to only a fraction of the available sites for hydrogen trapping. Hydrogen absorption is inhibited by the limitations of interstitial diffusion as presented by the disordered lattice. Retention of hydrogen at the subsurface layers of the alloy allows for saturation of the near-surface region, which in turn mitigates the transport of hydrogen atoms into the bulk leaving traps deeper in the material unfilled, or filled only partially. The absorption may be unfeasible in some local areas if the local solution enthalpy surpasses the value of the local activation energy for absorption, as illustrated in Figure 2. The effect of limited hydrogen absorption was observed in a previous study of isotopic exchange in the WMoTaNbV alloy where the effectiveness of deuterium removal by hydrogen gas was found to be negligible in the bulk of the material [19]. The study by Vuoriheimo et al. demonstrates the replacement of isotopes in gas heating taking place; heavy isotopes are replaced with lighter ones [19]. This indicates an identical trapping mechanism between the two isotopes. Should the trapping mechanism differ, even partially, the quantity of the implanted isotope would not reduce in gas heating as the absorbed isotope would take on different trapping sites. This hypothesis was verified in this work by comparing the total hydrogen isotope amounts between the D-implanted and plain WMoTaNbV alloy samples. The results are displayed in the Figure 3. Moreover, in Ref. [35], the statistical nature of the mechanism of the hydrogen isotope exchange is shown: the exchange takes place regardless of the hydrogen isotope as long as the trap has full hydrogen occupancy and a fast exchange can take place with the surrounding hydrogen in the lattice.

The above mechanisms prevent hydrogen concentrations from reaching the highest possible H/M ratios at room temperature, but also help retain much of the hydrogen once it is absorbed, as found in Ref. [18]. The limited diffusion due to local strains and chemical diversity may add to the energy barrier created by a primary trap, making its de-trapping energy higher. The complexity of the lattice is likely to have an effect on the activation energy as well, with the experimentally determined result $E_{A}=0.22\pm 0.02$ eV being effectively an average of a wider range of activation energies. The average of the experimental results hides the information on the magnitude of the activation energy scope. However, the obtained energy for the hydrogen solution fits well with the theoretically obtained values. The similar magnitudes between hydrogen absorption and the solution enthalpy suggest a low surface-to-bulk energy barrier in the eWMoTaNbV alloy. The absorption may still occur due to the chemical composition of the alloy enhanced by the hydrogen absorbing components V, Nb, and Ta [6,10,17], but is likely influenced by the microstructure as well. The previous research on hydrogen in bcc metals indicates an enhanced diffusion and accumulation of hydrogen to plastic deformation and zones with a high strain field, such as crack tips [36]. Thus, irregularities such as grain boundaries, voids, and dislocations may act as absorption sites for hydrogen.

## 6. Summary

We studied the dynamics of trapping hydrogen isotopes by determining the activation energy for solubility and the solution enthalpy of H in the WMoTaNbV alloy. The solution activation energy $E_{A}$ of H is experimentally determined to be $E_{A}=0.22\pm 0.02$ eV (21.2 ± 1.9 kJ/mol). Density functional theory calculations show that the neighbouring host atoms strongly influences the solution energy, leading to a range of values from −0.40 eV to 0.29 eV (−38.6 kJ/mol–28.0 kJ/mol). The study demonstrates that the complexity of lattice structure and elemental composition leads to a local energy variation in hydrogen trapping. The experimentally determined activation energy suggests low of average surface-to-bulk energy barrier, suggesting the absorption occurring due to chemical composition or the microstructure of the material.

## Figures and Tables

**Figure 1 materials-17-02574-f001:**
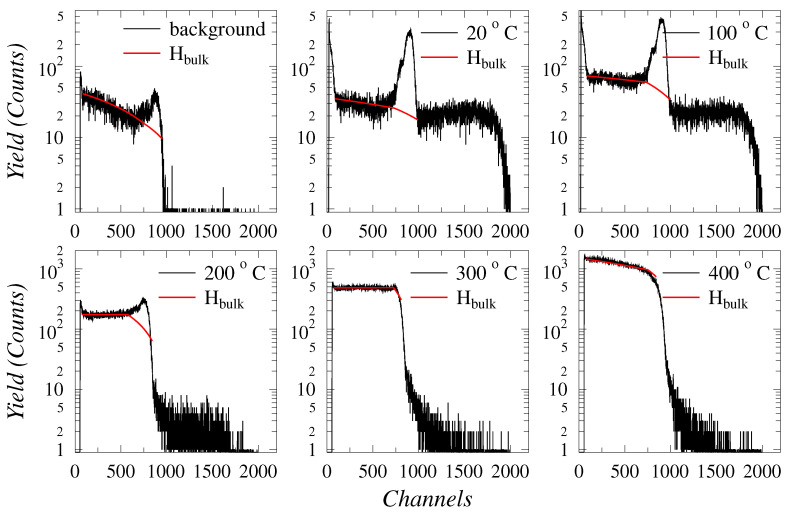
ERDA spectra of the raw data showing the retained H and D concentrations in the bulk as a function of the annealing temperature. Channels 0–1000: H concentration in the bulk, channels 1000–1750: retained D concentration. The H surface concentration was not considered in the analyses and was excluded from the ERDA statistics (seen as surface area between ERDA data (black line) and the H bulk data (red).

**Figure 2 materials-17-02574-f002:**
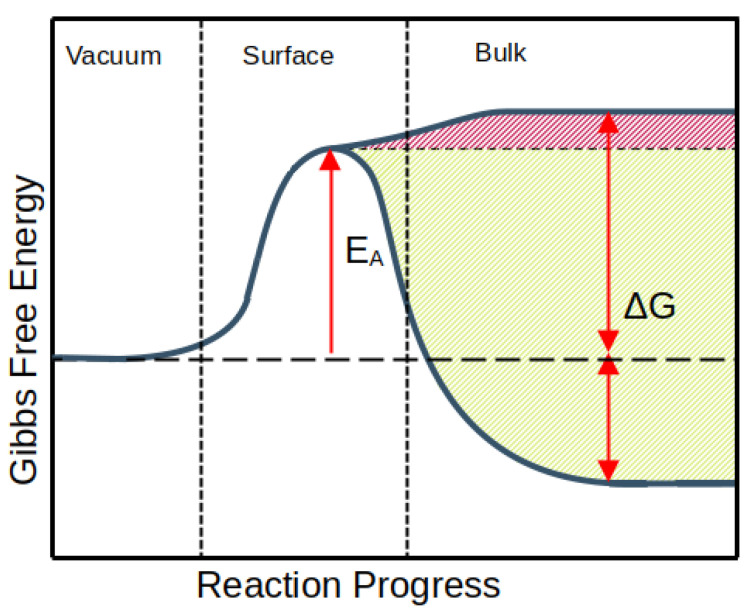
Illustration of the dynamics of the hydrogen absorption process. Hydrogen molecules in the vacuum are adsorbed on the material surface where the dimers are dissociated. Atomistic surface hydrogen must overcome a potential barrier described by an activation energy $E_{A}$ in order to penetrate the material’s subsurface and bulk regions. Depending on the material’s lattice structure and on the elemental configuration, the resulted solution energy ΔG can have a range of positive (endothermic reaction) or negative (exothermic) values.

**Figure 3 materials-17-02574-f003:**
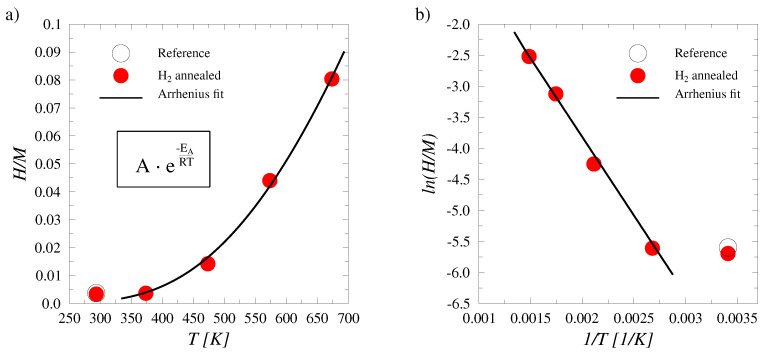
Fitting of the Arrhenius law to the equilibrium concentrations of hydrogen isotopes (H and D) as obtained with ERDA (statistical error in the measurements is less than 1.4%) The red data points mark the combined near-surface areal densities of hydrogen isotopes in the samples annealed in $H_{2}$ gas with temperatures up to 400 °C. H and D occupy the same trapping sites in the lattice of WMoTaNbV. The Arrhenius fit gives values of 0.22±0.02 eV for the activation energy and 1.3±0.3 for the pre-exponential factor. The goodness of fit, $R^{2}=$ 0.997. (**a**) The hydrogen isotope ratios to metal atoms as measured by ERDA as a function of gas annealing temperature. (**b**) The fit of the Arrhenius law.

**Figure 4 materials-17-02574-f004:**
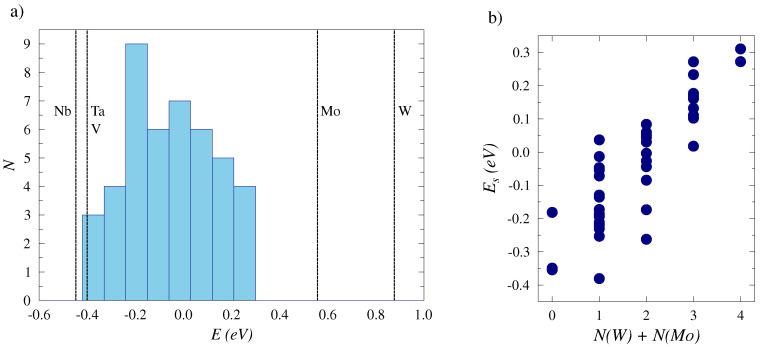
(**a**) The range of H solution enthalpies for the WMoTaNbV alloy and its constituents, as calculated with DFT. (**b**) The solution enthalpies as a function of Mo and W as the nearest neighbors for hydrogen.

## Data Availability

The data supporting this study are available from the corresponding author upon request. The data are not publicly available due to not being stored in any publicly archived datasets.

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
