# Peer review of "Solubility of Hydrogen in a WMoTaNbV High-Entropy Alloy"

_materials, 2024, doi:10.3390/ma17112574_

Round 1

Reviewer 1 Report

Comments and Suggestions for Authors

This article discusses a study on the dynamics of absorption of hydrogen isotopes by determining the activation energy for solubility and solution enthalpy of H in a WMoTaNbV high entropy alloy. They report that the solution activation energy of H is ~0.22 eV. A decent article although I have a few comments.

1. Please provide references for the listed equations, such as Sievert's law.

2. The two graphs in Figure 3 should be labelled (a) and (b).

3. Figure 4 is also missing the same labels.

4. Please provide an R2 value to the fitted line in Figure 3.

5. The conclusions of the study need to be elaborated upon in the Summary section.

Author Response

We thank the reviewer for reading the manuscript and providing constructive feedback.

  1. The references for the equations are now provided.
  2. The two graphs in the Figure 3 are now labelled (a) and (b).
  3. Labels (a) and (b) added to Figure 4 also.
  4. The R2 value (R2 = 0.997) is added in the caption of the Figure 3.
  5. The conclusions of the study are now elaborated upon in the Summary section with the addition of "The study demonstrates that the complexity of lattice structure and elemental composition leads to a local energy variation in hydrogen trapping. The experimentally determined activation energy suggests low of average surface-to-bulk energy barrier suggesting the absorption occurring due to chemical composition or the microstructure of material."

Reviewer 2 Report

Comments and Suggestions for Authors

In its current state, the manuscript can be reconsidered after significant revision.

Authors are strongly advised to concentrate on the following aspects for a significant revision. This answer is crucial to ensure that the manuscript meets the necessary standards for acceptance.

1. The introduction section presents deficiencies. The last paragraph of this section has no reason to exist; it is a partial copy of the abstract. Furthermore, it is necessary to add information that justifies why deuterium implantations were added to the alloy and hydrogen storage was evaluated. In addition, the justification and contribution of the research carried out to science are required. 

2. The method by which the chemical composition of the studied alloy was obtained, in addition to characterization results that justify the BCC  crystalline structure in the system studied, must be added.

3. On line 70, it is mentioned that an annealing treatment was carried out to eliminate impurities from the alloy. ¿What impurities is the author referring to? ¿The elements raw were not of high purity?

4. In Figure 1, the value on the y-axis in all graphs must be clarified and changed.

5. The equations of the enthalpy solution (2), Sievert's law (2), and Gibbs free energy ΔG°(3) must be referenced.

6. For Figure Caption 1, the text "For WMoTaNbV, the activation energy is determined experimentally resulting in a value EA = 0.22 ± 0.02 eV while the DFT calculated values for solution energy ΔG showed a distribution from −0.40 to 0.29 eV." It must be removed.

7. The distribution shown in Figure 4a extends from -40 to 40 eV, not "−0.40 eV with a distribution that stretches to 0.29 eV," as the text mentions.

8. In Figure 4, sections a and b must be added to each figure.

9. The discussion of the interaction mechanisms of hydrogen and deuterium and the effect of deuterium on the storage of H should be more in-depth.

10.- The title does not reflect the research carried out; since the indication that the samples were implanted with deuterium is omitted, it should be added.

11. Discuss the activation energies obtained, as was done for the solution enthalpy.

12. The conclusions must be rewritten; it is a copy of the abstract and the last paragraph of the introduction.

Author Response

We thank the reviewer for reading the manuscript and providing constructive feedback.

  1. The last paragraph of the introduction section has now been modified as follows. We expanded the previous work done, that formed the basis for this work providing a better justification for the carried out research. The deuterium implantations were made to study isotopic exchange combined with studying the effect of gas heating on the D distribution. Additionally, with the implantations we can verify the similarity between trapping mechanism of the isotopes. This is elaborated upon in the section of sample preparation and we have now provided more details in the discussion section (added part is in answer to question 9. of the referee).
  2. The method by which the chemical composition of the studied alloy was obtained was energy dispersive spectroscopy (EDS) and the characterization justifying the BCC crystalline structure was x-ray diffraction (XRD). This is now added to the manuscript.
  3. The annealing of the samples prior to gas heating is important to reduce levels of impurity atoms. By impurity atoms we refer to C, N, O and H of which the two former ones may be introduced in the casting process. This is now elaborated upon in the manuscript with the addition of " Casting of the alloy may introduce impurities of C, O and N in the lattice. In the prolonged high-temperature annealing N, O and H are desorbed along with partial removal of C through desorption as CO and CO2." in the section describing the sample preparation.
  4. The label of y-axis of Figure 1 is changed from YIELD to YIELD (COUNTS) for clarification. The values of the axis have not been modified as scaling them all equal will make it more difficult for the reader to see details of the data-analysis carried out (i.e. removal of the surface signal).
  5. The equations for Sievert's law and Gibbs free energy are now referenced.
  6. The text indicated by reviewer has been removed from caption of the Figure 1.
  7. The values given in the text are the correct ones. The graph has been corrected.
  8. In Figure 4, sections a) and b) are added to each figure.
  9. The following text has been added to the discussion section: "The study by Vuoriheimo et. al. demonstrates the replacement of isotopes in gas heating taking place; heavy isotopes are replaced with lighter ones [17]. This indicates identical trapping mechanism between the two isotopes. Should the trapping mechanism differ, even partially, the quantity of the implanted isotope would not reduce in gas heating as the absorbed isotope would take on a different trapping sites. This hypothesis was verified in this work by comparing total hydrogen isotope amounts between D implanted and plain WMoTaNbV alloy sample. The results are displayed in the Figure 3. Moreover, in Ref. [33] the statistical nature of the mechanism of hydrogen isotope exchange was shown: the exchange takes place regardless of the hydrogen isotope as long as the trap has full hydrogen occupancy and a fast exchange can take place with the surrounding hydrogen in the lattice."
  10. The authors consider the title reflecting the properly the research carried in this work. As is shown in this work, the implantations do not have an effect on the total concentration of D+H. This is verified with the comparison of total hydrogen amounts between implanted and unimplanted sample in the Figure 3.
  11. The activation energies are now discussed with the addition of text: "The averaging of the experimental results hides the information on the magnitude of the activation energy scope. However, the obtained energy for hydrogen solution fits well with the theoretically obtained values. The similar magnitudes between hydrogen absorption and the solution enthalpy suggests a low surface-to-bulk energy barrier in WMoTaNbV alloy. The absorption may still occur due to chemical composition of the alloy enhanced by the hydrogen absorbing components V, Nb and Ta [6,8,15], but is likely influenced by the microstructure as well. The previous research on hydrogen in bcc metals indicates an enhanced diffusion and accumulation of hydrogen to plastic deformation and zones with high strain field, such as crack tips [33]. Thus, the irregularities such as grain boundaries, voids and dislocations may act as an absorption sites for hydrogen."
  12.  The last paragraph of the introduction section was changed to reflect on the motivation for this work. After modifications the last paragraph elaborates on the previous and current research. The abstract was modified as well, by specifying the numerical values for the theoretical part of the work. Additionally, a short description of final conclusions has been included in the summary section. The repetition between sections was minimized.

Reviewer 3 Report

Comments and Suggestions for Authors

The manuscript is well-written and contributes to a better understanding of the hydrogen absorption in the WMoTaNbV high-entropy alloy. The methodology considers an experimental approach and density functional theory (DFT) calculations. The discussion of the results is scientifically strong. However, there is an important point that should be considered for clarification and/or revision.

Major comment:

[1] Although Reference 16 is cited (on page 2, line 64) to indicate where a detailed description of the microstructure of the WMoTaNbV alloy under study can be found, the manuscript should also provide a description of the microstructural characteristics of the analyzed samples. Furthermore, the influence of these microstructural characteristics should be further commented.

Author Response

We thank the reviewer for reading the manuscript and providing constructive feedback.

We have now described the details of the alloys microstructure in sample preparation section as follows:

"The casting process resulted in the alloy with a single bcc crystalline phase. The alloy had a fine grain structure, with the average grain size in order of ∼80um, although with significant variation between the sizes of individual grains [19]. Additionally, the gains had a dendritic shape and their elemental composition showed a microscale segregation of the constituent elements. The grain boundaries were enriched with Mo, V, and Nb and the centers of the grain with W [20]."

In the same section a statement regarding the effect of annealing temperature on the sample microstructure has been added:

"The annealing temperature was sufficiently low to not induce recrystallization in the samples [19]".

Additionally the effects of microstructure on hydrogen absorption have been considered in the discussion section:

"The similar magnitudes between hydrogen absorption and the solution enthalpy suggests a low surface-to-bulk energy barrier in WMoTaNbV alloy. The absorption may still occur due to chemical composition of the alloy enhanced by the hydrogen absorbing components V, Nb and Ta [6,8,15], but is likely influenced by the microstructure as well. The previous research on hydrogen in bcc metals indicates an enhanced diffusion and accumulation of hydrogen to plastic deformation and zones with high strain field, such as crack tips [34]. Thus, the irregularities such as grain boundaries, voids and dislocations may act as an absorption sites for hydrogen."

[34] Cui, T.; Dong, H.; Xu, X.; Ma, J.; Lu, Z.; Tang, Y.; Pan, D.; Lozano-Perez, S.; Shoji, T. Hydrogen-enhanced oxidation of ferrite phase in stainless steel cladding and the contribution to stress corrosion cracking in deaerated high temperature water. Journal of Nuclear Materials 2021, 557, 153209.

Reviewer 4 Report

Comments and Suggestions for Authors

The document is interesting and shows different results. Some adjustments must be considered to improve the quality of the document:

1. The abstract must contain some significant numerical results from the study and not just an introduction, to attract more readers and increase the rate of citation.

2. In the abstract use the word “enthalphy” must be “enthalpy”.

3. In the introduction section must be analyzed the previous state of the art including the work of the authors e.g [17] .

4. In the introduction section the authors use the word “interstitals” and must be “interstitials”.

5. In Fig. 1 use the word “spectras” for plural and must be “spectra” for plural and “spectrum” for singular.

6. “conjugent gradient energy” instead “conjugate gradient energy”

7. Please avoid unnecessary mistakes “which would indicate the the hydrogen absorption” and so it continues.

8. Please use a comprehensive English language review service.

9. Authors should present their experimental methods and parameters and sample processing and not only refer to other works. “The detailed description of the preparation process is available in Ref. [15]. The mechanical properties and microstructure of the material are outlined in Ref. [16]”.

10. The summary must be conclusions section that should include significant quantitative results of the study and not only qualitative results.

Comments on the Quality of English Language

The document is interesting and shows different results. Some adjustments must be considered to improve the quality of the document:

1. In the abstract use the word “enthalphy” must be “enthalpy”.

2. In the introduction section the authors use the word “interstitals” and must be “interstitials”.

3. In Fig. 1 use the word “spectras” for plural and must be “spectra” for plural and “spectrum” for singular.

4. “conjugent gradient energy” instead “conjugate gradient energy”

5. Please avoid unnecessary mistakes “which would indicate the the hydrogen absorption” and so it continues.

Author Response

We thank the reviewer for reading the manuscript and providing constructive feedback.

  1. Numerical results have been added to the abstract.
  2. In the abstract use the word “enthalphy” is now changed to “enthalpy”.
  3. The previous state of the art including the work of the authors e.g [17] has now been analyzed in the last paragraph of the introduction section: "The prospective for industrial applications of high-entropy alloys underline the importance of gaining qualitative knowledge about processes influencing absorption of hydrogen. In the previous research, a number of V-rich HEAs have been found absorbant of hydrogen, this includes the WMoTaNbV alloy studied in this work [15,16]. The WMoTaNbV efficiently traps hydrogen, with an energy of ∼1.7 eV [16]. The exchange between the H and D isotopes is however occurring at temperatures related to much lower energies, indicating a similarity between the trapping of two isotopes H and D [17]. This work confirms the hypothesis of the similar trapping mechanism between isotopes and investigates further the dynamics of hydrogen trapping. " As well as the discussion section with the addition of text: "The study by Vuoriheimo et. al. demonstrates the replacement of isotopes in gas heating taking place; heavy isotopes are replaced with lighter ones [17]. This indicates identical trapping mechanism between the two isotopes. Should the trapping mechanism differ, even partially, the quantity of the implanted isotope would not reduce in gas heating as the absorbed isotope would take on a different trapping sites. This hypothesis was verified in this work by comparing total hydrogen isotope amounts between D implanted and plain WMoTaNbV alloy sample. The results are displayed in the Figure 3. Moreover, in Ref. [33] the statistical nature of the mechanism of hydrogen isotope exchange was shown: the exchange takes place regardless of the hydrogen isotope as long as the trap has full hydrogen occupancy and a fast exchange can take place with the surrounding hydrogen in the lattice.
  4. In the introduction section the word “interstitals” has been changed to “interstitials”.
  5. In the Fig. 1 the word “spectras” has been changed to “spectra”.
  6. We have used the “conjugate gradient energy” to replace “conjugent gradient energy”.
  7. The unnecessary "the" has been removed and the manuscript checked for similar errors.
  8. The authors have now gone through the text carefully in terms of language and word selections in the manuscript and find the language being improved significantly and is of sufficient scientific quality.
  9. The experimental methods and parameters are now written out in the manuscript describing details of the alloys elemental composition and it's microstructure: "The casting process resulted in an alloy with a single bcc crystalline phase. The alloy had a fine grain structure, with the average grain size in order of ∼80µm, although with significant variation between the sizes of individual grains [19]. Additionally, the gains had a dendritic shape and their elemental composition showed a microscale segregation of the constituent elements. The grain boundaries were enriched with Mo, V, and Nb and the centers of the grain with W [20]." and "The concentrations of components in the alloy are reported to be Mo (20.7 at-%), Nb (20.3 at-%), Ta (20.4 at-%), V (19.2 at-%) and W (19.4 at-%) as determined by energy dispersive spectroscopy (EDS) and its bcc structure was confirmed by X-ray diffraction measurement (XRD) performed by manufacturer [18]."
  10. The final conclusions made on the basis of this work and the quantitative results of the study have been added to this section. The summary is left titled as summary.

Round 2

Reviewer 2 Report

Comments and Suggestions for Authors

After reviewing the version with the reviewer's recommendations, I consider that the modifications made in the manuscript contribute to the increase in the quality of work and meet the standards necessary for publication in the journal Materials.

Author Response

Thank you for your advice on the manuscript and for going through the changes made.

Reviewer 4 Report

Comments and Suggestions for Authors

The authors have made the corresponding corrections to the review comments. 

Author Response

(The authors gave the same response as above.)
